# Low Resilience Was a Risk Factor of Mental Health Problems during the COVID-19 Pandemic but Not in Individuals Exposed to COVID-19: A Cohort Study in Spanish Adult General Population

**DOI:** 10.3390/ijerph192215398

**Published:** 2022-11-21

**Authors:** Maria Llistosella, Pere Castellvi, Andrea Miranda-Mendizabal, Silvia Recoder, Ester Calbo, Marc Casajuana-Closas, David Leiva, Rumen Manolov, Nuria Matilla-Santander, Carlos G. Forero

**Affiliations:** 1Primary Health Care, Consorci Sanitari de Terrassa, 08227 Terrassa, Spain; 2Department of Nursing, Universitat International de Catalunya (UIC), 08195 Sant Cugat del Vallès, Spain; 3School of Medicine, International University of Catalonia (UIC), 08195 Sant Cugat del Vallès, Spain; 4Department of Basic Sciences, International University of Catalonia (UIC), 08195 Sant Cugat del Vallès, Spain; 5Servei Català de la Salut, 08028 Barcelona, Spain; 6Institut Universitari de Investigació en Atenció Primaria Jordi Gol (IDIAP Jordi Gol), 08007 Barcelona, Spain; 7Department of Social Psychology and Quantitative Psychology, University of Barcelona (UB), 08007 Barcelona, Spain; 8Unit of Occupational Medicine, The Institute of Environmental Medicine (IMM), Karolinska Institutet, 171 65 Stockholm, Sweden

**Keywords:** psychological resilience, COVID-19, major depressive disorder, anxiety disorders, suicidal ideation, post-traumatic stress disorders, mental health

## Abstract

Background: The aim is to analyze whether people with low resilience are at higher risk of mental health problems during the COVID-19 pandemic in Spanish adults. Methods: a longitudinal cohort study was carried out. Resilience was measured with the CD-RISC. Mental health problems that were assessed included: Major Depressive Episode (MDE), Generalized Anxiety Disorder (GAD), Suicidal Thoughts and Behaviors (STB), and Posttraumatic Stress Disorder (PTSD) symptoms. Results: we found statistically significant differences between groups and resilience scores in MDE [F (3; 48.40) = 19.55], GAD [F (3; 19.63) = 6.45] and STB [F (3; 111.74) = 31.94]. Multivariable analyses showed individuals with very low resilience were at a 5-fold risk of Incidence of MDE and a 4-fold risk of STB. Persistent group presented a 21-fold risk of MDE and 54-fold risk of STB. No evidence of higher risk was found for GAD. Individuals with low resilience and exposed to COVID-19 were not at higher risk. Individuals with low resilience were at higher risk of PTSD in general population [β(95% CI) = −3.25 (−3.969 to −2.54)], but not for individuals with COVID-19. Conclusions: in the general population, having low or very low resilience increases the risk of suffering MDE, STB, and PTSD, but not GAD during the COVID-19 pandemic, and not in the population with COVID-19.

## 1. Introduction

On 31 December 2019, the first case of the Coronavirus pandemic disease appeared (COVID-19) [1]. Most countries imposed a lockdown to prevent the spread and control the disease, such as Spain. The lockdown stipulated severe restrictions on social contact, working from home when possible, and significantly reduced access to services.

However, the COVID-19 pandemic had a psychological impact in the population. According to a meta-analysis, the prevalence of anxiety or depression was 30% [2], and in some countries doubled [3]. In Denmark [4] and the United Kingdom [5], results from a population-based cohort study showed worsening mental health. 

A cohort study carried out in the Spanish population showed that the prevalence of depression and suicidal ideation did not increase significantly in the spring and summer of 2020 [6]. However, it increased in Major Depressive Episode (MDE) (T0, 6.5% vs. T1, 8.8%) and Generalized Anxiety Disorder (GAD) (T0, 13.7% vs. T1, 17.7%) in the autumn of 2020 [7].

Mental health problems have also increased in the population affected by COVID-19. Two studies identified in a systematic review found that 96.2% of patients diagnosed by COVID-19 suffered posttraumatic stress disorder (PTSD) and 29.2% depressive symptoms [8]. Accordingly, another meta-analysis described that the most common symptoms among patients with SARS or MERS-CoV admitted to hospital were: depressed mood (6%) and anxiety (7%), and that in the post-illness stage, PTSD (32.2%), depression (14.9%), and anxiety (4.8%) [9]. Despite the psychological impact of the COVID-19 pandemic, some individuals did not develop psychological distress to be defined as resilient [10]. 

Resilience is defined as a psychological trait that enables the right coping with adversity [11]. This resilient trait has often been used to explain mental illness [12], hypothesizing that a resilient person can stay psychologically healthy even when exposed to harsh conditions in adverse environments. Resilience is a multidimensional trait comprising several domains, such as hardiness [13], positive emotionality [14], extraversion [15], self-efficacy [13], spirituality [16], self-esteem [17], and positive affect [12,18].

A review about the concept of resilience and mental health [11] concluded that resilience relates to rapid and effective recovery after stress [19]. Thus, it is considered an “immunological barrier” that allows a certain degree of health [20]. On the other hand, some authors associate the concept of resilience with promoting mental health [21]. Some of them even highlighted the importance of understanding resilience for developing interventions to prevent or treat mental disorders, particularly anxiety, depression, and stress [22].

Resilience and the resilient process in the population have been described in the literature, especially when dealing with traumatic events from natural disasters, such as the Haiti Earthquake in 2010 [23], in tsunami survivors [24], the Terrorist Attacks in 11 September 2001 in the United States [25]; or in the context of war [26].

From the COVID-19 pandemic, some research focused on the role of resilience and protective factors against stress and adverse outcomes derived from it, in the general population [27], psychiatric patients [28], or healthcare workers [29], among others. These studies showed that resilience is a protective factor for mental health; however, those studies used retrospective designs, such as cross-sectional data answered by participants or case-control studies. Thus, it is necessary to assess the role of resilience in the context of the COVID-19 pandemic for preventing and recovering from a mental health problem and designing effective mental health interventions based on promoting resilience.

## 2. Aims of the Study

The objective was to analyze whether people with low or very low resilience were at higher risk of mental health problems [MDE, GAD, suicidal thoughts, and behaviours (STB)] during the COVID-19 pandemic in the Spanish adult population and in people exposed to COVID-19 (diagnosed by COVID-19 or with related symptoms but no test undergone). To do so, the studied population was classified in four groups, depending on having or not mental health problems and their onset: (a) Healthy group (participants with no mental health problem assessed before and during the COVID-19 pandemic); (b) Recovery group (participants who recovered from any mental health problem appeared before COVID-19 pandemic); (c) Incident group (participants with a new-onset of any mental health problem during COVID-19 pandemic); and (d) Persistent group (participants with any mental health problem before and during COVID-19 pandemic). Finally, we assessed the association between resilience and PTSD symptoms in both populations during the COVID-19 pandemic.

## 3. Material and Methods

### 3.1. Study Design

A longitudinal population-based cohort study representative of the adult Spanish population in terms of geographical areas, sex, age, and socioeconomic characteristics was done. Two assessments were conducted using an online survey. The baseline reference survey (T0) was acquired as part of the BIOVAL-D study (ISCII-FEDER Exp: PI16/00165) during October/November 2019 and assessed mental health before the COVID-19 outbreak. The follow-up survey (T1) was conducted after 12 months (November/December 2020) using the same questionnaire and adding extra dimensions and variables to identify resilient factors associated with mental health problems and suicide risk after the COVID-19 outbreak and lockdown. The survey had an approximate duration of 30 min.

### 3.2. Participants

For baseline assessment (T0), participants were recruited from an online secure panel data vendor, including a final sample of 2005 individuals. The panel did not exclude any household and its members from the possibility of being recruited. Participants were randomly selected from an extensive list of mobile and landline phones as well as personal interviews, ensuring the panel’s representativeness. Participants received an informative email on the study objectives and characteristics, including a link to the informed consent, which will act as a filter for entering the survey. Participation was encouraged through an incentives system based on accumulative points and exchange. Points could be exchanged for physical and digital gifts or donations to NGOs. All participants who answered 100% of the survey accumulated the same number of points. Baseline participants were offered participation in the follow-up and were included upon signing informed consent. For the follow-up survey, all participants were recontacted by the panel provider and invited to participate also by email. 

From the initial 2005 individuals, 941 participants answered the follow-up survey, with a participation rate of 46.9%. An additional 416 subsamples, matched by sex, geographical residence, and age, were invited to participate to ensure population representativeness. The final sample summed a total of 1357 participants for 12-months analysis. 

### 3.3. Variables and Instruments

#### Resilience (T1)

We administrated the 10-items Connor-Davidson Resilience Scale (CD-RISC) [30], from the 25-items CD-RISC [11]. The CD-RISC is a self-administered questionnaire composed of 10 Likert-type items with five response options (0 = not true at all, 1 = rarely true, 2 = sometimes true, 3 = often true, and 4 = true nearly all of the time) [11]. The scale measures stress coping ability for use in clinical samples. The range of values of the item-total scale score correlation was 0.45 to 0.69, and Cronbach’s alpha was 0.85 [30]. The highest scores indicated the highest level of resilience [11]. 

### 3.4. Mental Health Problems

#### 3.4.1. Major Depressive Episode (MDE) (T0 & T1)

MDE diagnosis was based on the MDE screening section (8-items) of the Composite International Diagnostic Interview (CIDI) version 3.0 [31]. This section worked as a filter to enter the diagnostic section, which includes 37 symptoms divided into eight sections (depression and anhedonia, 6-items; weight, 5-items; insomnia, 5- items; retardation and agitation, 4-items; fatigue, 2-items; worthlessness and guilt, 5-items; concentration, 4-items; suicide, 6-items) that evaluate the presence or absence of MDE symptoms for at least two weeks during the past 12 months. Individuals with five active criteria must meet a disability over 50 points in the WHODAS score to establish the diagnostic [32]. The area under the curve was 0.75 [33]. The CIDI has been translated into many languages, including Spanish [31].

#### 3.4.2. Generalized Anxiety Disorder (GAD) (T0 & T1) 

The Generalized Anxiety Disorder-7 scale (GAD-7) was administered. The GAD7 consists of 7 items answered with a 4-point Likert scale from 0 to 21. Considering the 10-point cut-off, sensitivity values of 86.8% and specificity of 93.4% were found [34]. For the Spanish version, Cronbach’s alpha coefficient was 0.93.

#### 3.4.3. Suicidal Thoughts and Behaviours (STB) (T0 & T1)

Any STB was assessed for ideation, plan or attempt with a single item for each symptom from the CIDI questionnaire [35]. Suicidal ideation is classified as passive and active ideation, and suicidal and suicide attempt. Response items were Yes/No.

MDE, GAD, and STB categorized as (a) Healthy (negative in T0 and T1); (b) Incidence (negative in T0 but positive in T1); (c) Persistence (positive T0 and and T1) and (d) Recovery (positive in T0 and negative in T1) groups.

#### 3.4.4. Post-Traumatic Stress Disorder (PTSD) Symptoms (T1)

To assess DSM-5 symptoms of PTSD related to the experience of being infected with COVID-19 or the death of somebody close due to COVID-19, an adapted Spanish version of the PTSD Checklist for DSM-5 (PCL-5) was used (20 items) [36]. 

The PCL-5 was adapted ad hoc in the context of being exposed to the COVID-19 pandemic (e.g., …avoid memories, thoughts, or feelings related to being infected or someone has died from COVID-19).

The PCL-5 was administered to participants who underwent a COVID-19 diagnostic test, all those who knew a person who died from COVID-19, and a randomized selection of 20% of the rest of the sample [7].

#### 3.4.5. COVID-19 Exposures (T1)

Items about COVID-19 exposure were developed ad hoc for this study [37]. Data about having been tested for or diagnosed with COVID-19 were gathered, including related symptoms. We considered four groups of COVID-19 clinical status: (a) Group 1, individuals with no COVID-19 test done and no COVID-19 symptoms; (b) Group 2, those with COVID-19 symptoms but no test done; (c) Group 3, those with a negative COVID-19 test result; and (d) Group 4, those with a positive COVID-19 test result.

#### 3.4.6. Demographic Variables

Age, gender, employment status, and marital status were assessed. These variables were considered potential confounders.

## 4. Data Analysis

According to Considering CD-RISC Manual [38], we considered four categories of resilience using quartiles (*Q*_1_ = very low resilience; *Q*_2_ = low resilience; *Q*_3_ = high resilience; *Q*_4_ = very high resilience). 

Descriptive statistics for continuous variables were means and standard deviations and absolute frequencies and percentages for categorical variables. 

To test for differences between groups (Healthy; Recovery; Incidence and Persistence) in mental health problems (MDE, GAD, and STB) and resilience scores (continuous), we used an analysis of variance (ANOVA). Homogeneity of variance was met according to Levene’s test. Post-hoc analyses were conducted with Tukey’s test, using Cohen’s *d* to measure affect size.

We estimated the bivariate and adjusted associations among resilience categories and other variables using crude (OR) and adjusted Odds Ratios (AOR). We estimated resilience association with the 12-month presence of MDE, GAD, and STB in the total sample and in the subgroup of COVID-19 exposed participants (COVID-19 test positives or with COVID-19 compatible symptoms). Logistic regression adjusted ORs controlling for potential confounder variables: age, gender, employment status, and marital status. Subgroup analysis among the COVID-19 exposed group was done, collapsing confounder categories due to the low number of individuals in each group. 

Finally, the association between resilience and PTSD symptoms was estimated with non-parametric tests, as homogeneity hypotheses did not hold (*p* < 0.05). Spearman’s correlations were used with continuous resilience scores and the Kruskal-Wallis test when resilience was treated as categorical. Non-parametric tests were used. Bivariate and multivariable linear regression models (adjusted by age, gender, employment status, and marital status) were used for assessing the relationship between PTSD and resilience in the total sample and individuals exposed to COVID-19.

All analyses used complex sample design weights. Due to non-random losses to follow-up, selection bias was corrected using inverse probability-of-censoring post-stratification weights [39]. 

All statistical tests were conducted with R package *ipw* [40] and SPSS v20.0 with a nominal significance level of 5%.

### Ethics and Dissemination

The Ethical Committee approved the study of the International University of Catalonia (UIC) (ref: MED-2020-02). The previous study was approved by the Ethical Committee of IMIM-Parc de Salut Mar (ref: 2014/4714/I).

Before entering the study, participants read and signed informed consent. Data were gathered in an entirely anonymous approach, with no possibility to match the data with the participant’s identity.

## 5. Results

The average resilience score of participants in the follow–up was 24.50 (*SD* = 6.93), and the median (*Q*_1_–*Q*_3_) was 25 (20–29). 

Table 1 shows sample descriptives: 48.5% were men, 53.8% of the sample had an age range of 40–65 years, 31.3% were single, 56.6% were married, more than half of the sample were employed (54.6%), and 20.4% were retired; 514 participants (38.1%) of the total sample received a diagnostic test for COVID-19, and 3.4% reported symptoms related to COVID-19, but they did not undergo any diagnostic test. Of the 514 participants who had a COVID-19 diagnostic test, 48 (9.3%) were positive, representing 3.6% of the total sample. The percentages of individuals with MDE, GAD, and STB were, respectively: 6.5%, 13.7%, and 15.1% at baseline, and 8.8%, 17.7%, and 7.1%, at 12- months of follow-up. The total sample’s median (*Q*_1_–*Q*_3_) of PTSD symptoms was 14 (7–34). 

Resilience total score differences between the Incidence, Persistence, Recovery and Healthy groups in MDE, GAD and STB are reported in Figure 1. Statistically significant differences were found between resilience and all groups of mental health problems: MDE (F [3; 1854] = 41.74, η^2^ = 0.063, *p* < 0.001), GAD (F [3; 309] = 17.02, η^2^ = 0.142, *p* < 0.001) and STB (F [3; 1774] = 28.47, η^2^ = 0.046, *p* < 0.001).

With respect to the healthy group, post-hoc analyses showed statistically significant differences with all three groups in MDE (Incidence, Persistence, Recovery) (*p* < 0.001). In GAD and STB, statistically significant differences were found between Healthy group and Persistence (GAD and STB, *p* < 0.001) and Recovery (GAD, *p* = 0.001; STB, 0.035) groups, but not with Incidence group in GAD (*p* = 0.380) and STB (*p* = 0.133). Comparing groups for each mental health problem, the Healthy group had the highest score of resilience and Persistence group the lowest regarding MDE (Healthy, mean = 25.03, *SD* = 6.98; Incidence, mean = 20.31, *SD* = 6.88, *d* = 0.683; Persistence, mean = 16.82, *SD* = 8.08, *d* = 1.186; Recovery, mean =20.34, *SD* = 6.25, *d* = 0.678) and STB (Healthy, mean = 24.99, *SD* = 7.10; Incidence, mean = 22.43, *SD* = 6.30, *d* = 0.369; Persistence, mean= 17.66, *SD* = 6.56, *d* = 1.057; Recovery, mean = 23.33, *SD* = 6.79, *d* = 0.239). As for GAD, the highest score of resilience were found for Recovery group and the lowest for Persistence group (Healthy, mean= 21.52, *SD* = 6.06; Incidence, mean = 20.15, *SD* = 5.79, *d* = 0.197; Persistence, mean= 13.88, *SD* = 6.45, *d* = 1.102; Recovery, mean = 26.85, *SD*= 5.28, *d* = −0.770).

Table 2 shows crude and adjusted OR values between resilience and a disorder in the incidence, persistence, and recovery groups. In the general population, logistic regression results show that very low resilience (*Q*_1_) increased the risk of suffering MDE and STB in all three groups compared with those with very high resilience (*Q*_4_), both in crude and adjusted analyses. Conversely, this was not the case for GAD, where participants with low resilience (*Q*_2_) also increase the risk of suffering MDE and STB, in the Persistence and Recovery group for STB and Incidence group for MDE. There was a statistically significant higher risk of MDE in individuals with very low resilience (*Q*_1_) comparing the Healthy group with the Incidence group: bivariate, OR 5.16; 95% CI 2.88–9.23; multivariable, OR 5.47; 95% CI 2.92–10.26; with the Persistence group: bivariate, OR 18.56; 95% CI 5.21–66.07; multivariable, OR 21.17; 95% CI 5.16–86.79; and with the Recovery group: bivariate, OR 5.98; 95% CI 2.84–12.55; multivariable, OR 5.65; 95% CI 2.59–12.33, respectively.

Additionally, the association between resilience and MDE in individuals with low resilience (*Q*_2_) was statistically significant for Incident group (bivariate, OR 2.32; 95% CI 1.24–4.35; multivariable, OR 2.51; 95% CI 1.29–4.89), Persistent group (bivariate, OR 4.89; 95% 1.24–19.22), and Recovery group (multivariable, OR 2.44; 95% 1.03–5.76). There was a statistically significantly higher risk of STB in individuals with very low resilience (*Q*_1_) when comparing the Healthy group with the Incidence group: bivariate, OR 3.13; 95% CI 1.20–8.19, multivariable, OR 4.14; 95% CI 1.47–11.62; with the Persistence group: bivariate, OR 54.35; 95% CI 10.44–282.92; multivariable, OR 53.92; 95% CI 10.21–284.87; and with the Recovery group: OR 2.43; 95% CI 1.48–3.998; multivariable, OR 2.11; 95% CI 1.27–3.51. In individuals with low resilience (*Q*_2_), there was a statistically significantly higher risk of STB comparing the Healthy group than in the Persistence and Recovery group. The OR values were: to the Persistence group: bivariate, OR 14.57; 95% CI 2.66–79.78; multivariable, OR 14.38; 95% CI 2.59–79.86; and to the Recovery group: OR 1.99; 95% CI 1.21–3.25; multivariable, OR 1.95; 95% CI 1.17–3.23. Furthermore, in individuals with high resilience (*Q*_3_), there was a statistically significantly higher risk of STB comparing the Healthy group than in the Recovery group in both models (bivariate: OR 9.83; 95% CI 1.76–54.83; multivariable: OR 10.86; 95% CI 1.91–61.59) In individuals with very low (*Q*_1_) and low resilience (*Q*_2_), the association between resilience and GAD was only statistically significant for the Recovery group in the bivariate model. The OR values were: (*Q*_1_) OR 0.14; 95% CI 0.03–0.79 and (*Q*_2_) OR 0.14; 95% CI 0.02–0.86. Multivariable analyses for Recovery and Persistence groups and bivariate analysis for Persistence groups could not be estimated for GAD.

Table 3 shows crude and adjusted ORs from bivariate and multivariable models in the COVID-19 exposed population. No significant associations (*p* ≥ 0.05) were found for GAD and STB when comparing Healthy or Recovery groups with Incidence or Persistence groups when adjusted by age, gender, employment status, and marital status. As for MDE, a statistically significant adjusted association (*p* = 0.046) was found in individuals with low resilience.

Finally, results show a statistically significant correlation between PTSD symptoms and resilience scores (continuous) (rho= −0.301; *p* < 0.001), as well as group differences between PTSD symptoms and resilience using quartiles (categorical) in the general population (*p* < 0.001). When we analyzed among those individuals exposed to COVID-19, we found a statistically significant Spearman correlation (rho= −0.257; *p* = 0.027) but not when resilience was categorical (*p* = 0.116) (see Figure 2). Bivariate linear regression using resilience showed a statistically significant association [β(95% CI) = −3.46 (−4.19–−2.74); *p* < 0.001] when it was adjusted by age, gender, employment status, and marital status [β(95% CI) = −3.25 (−3.969–−2.54); *p* < 0.001] in the total population. Among individuals exposed to COVID-19, there was not an association in either the bivariate [β(95% CI) = −3.17 (−6.64–−0.30); *p* = 0.073], nor in the multivariable model [β(95% CI) = −3.18 (−6.44–0.087); *p* = 0.056].

## 6. Discussion

### 6.1. Main Results

The present study investigates whether people with lower resilience implied a higher risk of suffering mental health problems during the COVID-19 pandemic in the Spanish adult population. Four groups were analyzed depending on whether or not they had any of the mental health problems evaluated before or during COVID-19 pandemic: (a) Healthy group (participants with no mental health problem in both assessments); (b) Recovery group (participants who recovered during from any mental health problem at baseline); (c) Incident group (participants with a new-onset of any mental health problem at follow-up); and (d) Persistent group (participants with any mental health problem in both assessments). 

As expected, individuals with very low and low resilience showed a higher risk of having any mental health problem than those with high resilience during the COVID-19. This effect was found for Major Depressive Episode (MDE) and suicidal thoughts and behaviors (STB) independent of age, gender, employment status, and marital status. Results showed a gradual increase in the risk of suffering MDE or STB depending on the degree of resilience. Our results also showed higher resilience in the Healthy and Recovery groups, except for MDE, where the Recovery group had a similar risk to the Incident group. Incident and Persistent groups, which were active cases of mental health problems during the pandemic, showed lower resilience. The highest risk was found for the Persistence group, with a 21-fold risk of MDE and 54-fold risk of STB, respectively. However, the GAD group with the highest resilience was the Recovery group. When we analyzed resilience in the COVID-19 exposed subgroup, there was no association with any mental health problem analyzed. The PTSD symptoms and resilience are associated in the general population but not in individuals exposed to COVID-19.

### 6.2. Comparison with Other Studies

In the total population, having very low resilience increases the risk of MDE or STB. These results are consistent with the literature in risk contexts [41,42,43]. Resilience has been related to psychological well-being in multiple studies [11,18], especially with decreased depressive and anxious symptoms [41]. Killgore et al. [42] found that fewer resilience levels during the COVID-19 pandemic were associated with more severe mental health problems in the general population.

Individuals with very low resilience were at a 5-fold greater risk of a new-onset of an MDE and 4-fold risk of STB than those with high resilience. This result makes us consider the Incident group the most specifically affected by the pandemic. However, the Persistence group, who had a mental health problem before the pandemic, were especially at-risk with a 21-fold risk of an MDE and 54-fold risk of STB, respectively.

Against what we expected, we did not find a statistically increased risk of suffering GAD in the low resilience groups. However, the low number of individuals in each group may be related to low power to detect the relationships in multivariable regressions. Also, we have only measured resilience from an individual perspective. Other environmental factors such as family or peer support, which are protective from anxiety [27,43], have not been considered.

The study results by Havnen et al. [27] showed that resilience moderates the relationship between stress (suffered by the COVID-19 pandemic) and anxiety symptoms. Resilience was assessed with the questionnaire Resilience Scale for Adults (RSA), which includes not only individual factors, but also family and social ones [44].

A study in the Italian population [10] found specific factors that hindered resilience during the COVID-19 pandemic, such as anxiety and PTSD symptoms related to COVID-19. However, beyond infection itself, isolation periods with interruption of usual routines and diminished social contacts, such as the lockdown or loss of employment, are known significant stressors. Another previous study showed uncertainty, loneliness, living with children, higher education, and living in regions where the virus was beginning to spread [10]. These results suggest that the implication of resilience in buffering the psychological impact can be established in the general population affected by the stressors of COVID-19.

Our results complement other studies in which different factors have been identified to influence the psychological response to the pandemic. A review of the literature conducted by Gilan et al. [45] revealed an increase in stress in the general population with manifestations of depression, anxiety, PTSD, and sleep disturbance. The risk factors identified were patient contact, female sex, impaired health status, worry about family members and significant others, and sleep quality. Contrarily, being informed about the number of individuals who have recovered from COVID, social support and perceived infectious risk were identified as resilient factors [45]. Another study revealed that older age and employed status contributed to decreased symptoms of anxiety [27].

### 6.3. COVID-19 Subgroup

Unexpectedly, individuals with low or very low resilience, and who were exposed to COVID-19 (tested positive or have shown COVID-19-related symptoms), were not at higher risk of mental health problems. These results could be explained by the low power of the multivariable statistical models with the involved sample sizes. However, other explanations would be worth exploring. For instance, the action of SARS-CoV-2 has on the brain, which may be attenuating the protective effects of resilience. In fact, COVID-19 has short- and long-term neuropsychiatric symptoms and long-term brain sequelae, showing anosmia, cognitive and attention deficits (i.e., brain fog), new-onset anxiety, depression, psychosis, seizures, and even suicidal behavior, because SARS-CoV-2 can damage endothelial cells leading to inflammation, thrombi, and brain damage [46]. Another explanation would be the cumulative effect of risk factors that might cause an allostatic load that increases vulnerability to new onsets of mental health problems [47,48]. According to this, individuals with COVID-19 could be exposed to a higher number of typical stressors of the pandemic situation described above.

### 6.4. Implications for Clinical Practice

Immediate steps to increase mental health support were taken by countries. For instance, most countries implemented measures to protect jobs and incomes; job retention schemes helped to support both the incomes and the mental health of workers. In addition, phone support lines were developed to give tips on coping measures during the COVID-19 pandemic [3].

The main areas of mental health policy applied to some countries during the COVID-19 pandemic were legislation, regulation, financing, accountability, and workforce development [49]. However, there is broad support in the literature that it should be implement policies that act on the social determinants of health in mental health policies [49]. In the literature on resilience, the importance of the emotional and social dimensions of the lives and histories of individuals, families, organizations, and communities is evident [50].

As resilience probably mediates MDE, STB, and PTSD regarding the epidemic, these findings suggest that promoting resilience may be an essential target for effective psychological interventions for the general adult population; especially for the most vulnerable populations that experienced higher rates of mental distress during the COVID-19 pandemic such as people with less secure employment, lower educational level, and lower-income [3].

This research reinforces the need to design and implement programs to promote resilience from mental health policy, to increase the protective factors related to resilience, and to reduce the risk of suffering from mental problems such as MDE and STB, especially for future crises or pandemics.

### 6.5. Limitations & Strengths

The results of this study must be seen in light of its limitations. First, we only took into account resilience as a trait. External factors, such as social or familial support, are associated with resilience [51], ameliorating the impact of an adverse event. However, the Connor-Davis scale is a valid and reliable questionnaire with good psychometric properties [11]. Second, mental health problems and suicide risk assessment was based on self-reports and not on direct clinical assessment. Therefore, they should be considered as “probable cases” of disorder.

Nevertheless, an excellent diagnostic agreement has been reported with the clinical judgment of the CIDI instrument, which includes the assessment of MDE and STB in our study, with face-to-face [33] and online assessments [52] into the Spanish population. Also, we analyzed two mental disorders (MDE, GAD), STB, and symptoms of PTSD using adapted and validated instruments. Third, the diagnostic test for COVID-19 was self-reported, and there is a probability of it being false negative due to fear of telling the truth or the stigma related to having COVID-19. Fourth, the intervals of confidence in many of the results have been broad, so they must be interpreted cautiously. Finally, although mental health problems were assessed at baseline and follow-up, resilience was evaluated during the pandemic but not before. Ideally, resilience should have been assessed before the pandemic, but the aim of the previous study (baseline assessment) was not to assess the role of resilience in the prevention of mental health problems at the population level. Then, the resilience may be worsened among those with a mental health problem during the follow-up, showing lower resilience than if we assessed it before the pandemic. Therefore, the results should be considered with caution. However, the resilience construct should be considered as a trait and therefore with an appropriate construct stability over time during adulthood.

Nevertheless, the study has several strengths. To our best knowledge, this is the first prospective study assessing prevalent mental disorders before and during the first year of the COVID-19 pandemic. It has also been possible to explore resilience in the different health states that assess trajectories of these mental health problems and in people exposed to COVID-19. Second, we recruited a representative sample of the Spanish adult population, and the results could be generalizable to the rest of the Spaniards. Finally, the online methodology tends to deliver more reliable information about sensitive information, such as suicide risk, than face-to-face assessments [53].

## 7. Conclusions

The longitudinal design has allowed us to assess the trajectories of mental health problems (Healthy, Incidence, Persistence, Recovery) depending on resilience. Low resilience increases the risk of suffering a Major Depressive Episode, Suicidal Thoughts and Behaviours, and PostTraumatic Stress Disorder symptoms, but not the risk of suffering a Generalized Anxiety Disorder during the pandemic in the total population. These findings are consistent with previous research on resilience in adverse situations, including the COVID-19 pandemic, where resilience has emerged as a protective factor against mental problems in the population. However, no association was found in the population exposed to COVID-19 (diagnosed by or with symptoms of COVID-19), probably due to brain damage caused by the infection. Social policies aimed to protect and promote resilience may be adequate to improve the mental health of the most vulnerable populations.

## Figures and Tables

**Figure 1 ijerph-19-15398-f001:**
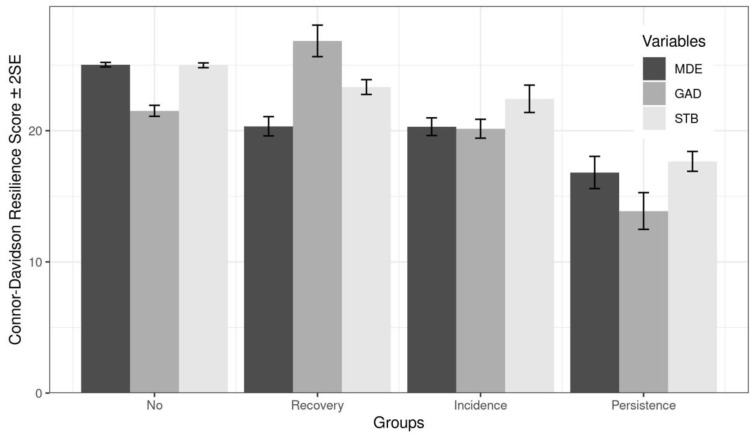
Resilience total score differences between four groups (Incidence, Persistence, Recovery, and Healthy), defined by MDE, GAD, and STB scores. Differences statistically significant (*p* < 0.001) between groups (No; Recovery; Incidence; Persistence) and resilience total score for all disorders.

**Figure 2 ijerph-19-15398-f002:**
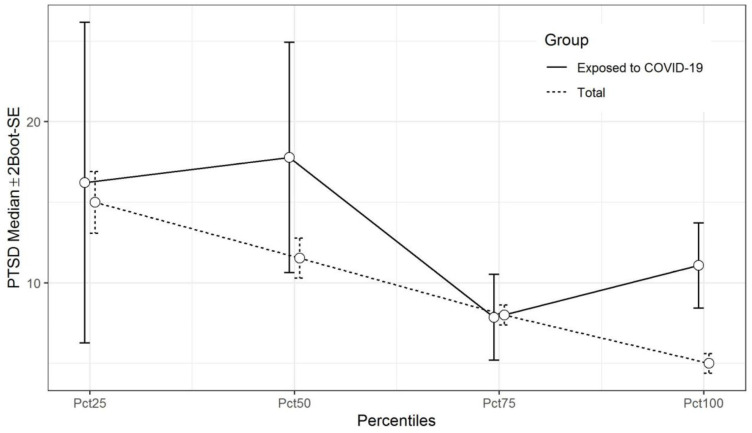
Differences of PTSD symptoms in the total sample and individuals exposed to COVID-19. Weighted medians and standard errors (estimated through a Bootstrap procedure with 5000 replications) of PTSD scores for all resilience percentile groups.

**Table 1 ijerph-19-15398-t001:** Characteristics of the sample characteristics at baseline and 12-month follow-up.

	Baseline		
	N = 2005		
	N	%	SE.
**Socio-demographic characteristics**			
Gender (men)	969	48.5	0.35
Age, (years)			0.18
18–25	125	6.3	
>25–40	444	22.2	
>40–65	1074	53.8	
>65	355	17.8	
Marital status			0.23
Single	625	31.3	
Married/Couple	1130	56.6	
Separated	23	1.1	
Divorced	120	6	
Widowed	100	5	
Employment status			0.06
Employed	1090	54.6	
Off sick	30	1.5	
Unemployed	158	7.9	
Homemaker	139	7	
Student	71	3.6	
Student & employed	57	2.8	
Temporal or permanent disability	44	2.2	
Retired	408	20.4	
**Mental health**			
Major depressive episode	131	6.5	0.01
Generalized anxiety disorder	99	13.7	0.01
Any suicidal thoughts and behaviors	294	15.1	0.01
Suicidal ideation (passive)			0.34
Yes	276	13.8	
I don’t know	50	2.5	
Suicidal ideation (active)			0.29
Yes	87	4.4	
I don’t know	34	1.7	
**Resilience** *(Mean (SD))*	24.5	−6.93	0.23

SD = Standard deviation; S.E. = Standard error. % weighted follow-up sample weight (inverse probability weighting and post-stratification).

**Table 2 ijerph-19-15398-t002:** Crude and Adjusted ORs of very low (*Q*_1_), low (*Q*_2_), and high resilience (*Q*_3_) versus very high resilience (*Q*_4_) and 12-month presence of a mental health problem (MDE, GAD, and STB) in the Incidence, Persistence and Recovery groups in the total sample.

		Disorder Course											
		Incidence				Persistence				Recovery			
		Bivariate		Multivariable		Bivariate		Multivariable		Bivariate		Multivariable	
Mental Health Problem	Resilience	Crude OR	95% CI	aOR	95% CI	Crude OR	95% CI	aOR	95% CI	Crude OR	95% CI	aOR	95% CI
Major Depressive Episode	Very high (ref)	--	--	--	--	--	--	--	--	--	--	--	--
	High	0.79	0.36–1.73	0.72	0.31–1.64	0.841	0.13–5.44	0.61	0.08–4.70	1.85	0.81–4.26	2.24	0.94–5.37
	Low	2.32	1.24–4.35 **	2.51	1.29–4.89 *	4.89	1.24–19.22 *	3.99	0.89–17.82	2.21	0.97–5.05	2.44	1.03–5.76 *
	Very low	5.16	2.88–9.23 **	5.47	2.92–10.26 **	18.56	5.21–66.07 **	21.17	5.16–86.79 **	5.98	2.84–12.55 **	5.66	2.60–12.34 **
Generalized Anxiety disorder	Very High (ref	--	--	--	--	--	--	--	--	--	--	--	--
	High	0.88	0.28–2.76	0.53	0.12–2.43	n.a	n.a	n.a	n.a	1.35	0.42–4.39	2.13	0.35–12.91
	Low	1.5	0.55–4.1	1.56	0.48–5.05	n.a	n.a	n.a	n.a	0.14	0.02–0. 86 *	0.14	0.01–1.45
	Very Low	1.57	0.59–4.23	1.54	0.49–4.80	n.a.	n.a.	n.a.	n.a.	0.14	0.03–0.79 *	0.18	0.02–1.53
Suicidal Thoughts and Behaviors	Very High (ref)	--	--	--	--	--	--	--	--	--	--	--	--
	High	1.74	0.64–4.75	2.36	0.82–6.84	9.83	1.76–54.83 *	10.86	1.91–61.59 *	1.31	0.78–2.19	1.24	0.73–2.11
	Low	1.79	0.64–5.01	1.81	0.61–5.34	14.57	2.66–79.78 *	14.38	2.59–79.86 *	1.99	1.21–3.25 *	1.95	1.17–3.23 *
	Very Low	3.13	1.20–8.19 *	4.14	1.47–11.62 *	54.35	10.44–282.92 **	53.92	10.21–284.87 **	2.43	1.48–3.998 **	2.11	1.27–3.51 *

aOR = adjusted odds ratio; GAD = Generalised Anxiety Disorder; MDE = Major Depressive Episode; n.a., not applicable; Ref., Reference group; STB = Suicidal Thoughts and Behaviors. Multivariable models adjusted by: Age, gender, employment status, and marital status. Resilience was categorized in quartiles: lower than 25th percentile (very low resilience); 25–50th percentile (low resilience); 50–75th percentile (high resilience) of our sample and than 75th (very high resilience) in our sample. * *p* < 0.05; ** *p* < 0.001.

**Table 3 ijerph-19-15398-t003:** Bivariate and multivariable associations of very low (*Q*_1_), low (*Q*_2_), and high (*Q*_3_) resilience versus very high resilience (*Q*_4_) with MDE, GAD, and STB at 12-month follow-up in patients in the population in people exposed to COVID-19, comparing those with Incidence or Persistence with Healthy or Recovery.

		Bivariate			Multivariable		
Presence of a Mental Health Problem (Incident or Persistent)	Resilience	Crude OR	95% CI	*p*-Value	aOR	95% CI	*p*-Value
Major Depressive Episode	Very high (ref)	--	--	--	--	--	--
	High	1.20	0.29–4.94	0.307	1.12	0.15–8.45	0.915
	Low	0.75	0.19–2.99	0.686	0.11	0.01–0.96	0.046 *
	Very low	1.86	0.57–6.12	0.307	2.08	0.45–9.64	0.349
Generalized Anxiety Disorder	Very High (ref)	--	--	--	--	--	--
	High	n.a	n.a	n.a	n.a	n.a	n.a
	Low	1.11	0.82–14.83	0.939	n.a	n.a	n.a
	Very Low	1.60	0.16–16.62	0.692	n.a	n.a	n.a
Suicidal Thoughts and Behaviors	Very High (ref)	--	--	--	--	--	--
	High	n.a.	n.a	n.a	n.a	n.a	n.a
	Low	0.85	0.22–3.32	0.812	0.36	0.06–2.28	0.359
	Very Low	1.15	0.31–4.31	0.833	1.68	0.33–8.42	0.531

GAD = Generalised Anxiety Disorder; MDE = Major Depressive Episode; n.a., not applicable; Ref., Reference group; STB = Suicidal Thoughts and Behaviors. Multivariable model adjusted by: Age, gender, employment status, and marital status. * *p* < 0.05.

## Data Availability

The data presented in this study are available on request from the corresponding author.

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
