# Peer review of "Low Resilience Was a Risk Factor of Mental Health Problems during the COVID-19 Pandemic but Not in Individuals Exposed to COVID-19: A Cohort Study in Spanish Adult General Population"

_ijerph, 2022, doi:10.3390/ijerph192215398_

Round 1

Reviewer 1 Report

This is an interesting paper. A few revisions.

Introduction:

Take care using the word "successful" as it can imply judgment. 

Methodology:

Reads clearly, but more information required about how participants were recruited and sampled. 

How were the four groups (healthy etc) defined?

Make sure to write "data were", not "data was".

Results: 

Clearly explained.

Discussion:

Make sure to define the four groups of the study to strengthen argument and connections. 

Limitations section should be put at the end of the discussion as it disrupts the flow of the discussion (comparison to other studies etc) with it in the middle. 

Author Response

Response to Reviewer 1 Comments

Point 1:  Introduction: Take care using the word "successful" as it can imply judgment. 

Response 1: Thank you very much for the contribution; We changed “successful” to “right” to avoid any misunderstanding.

Point 2: Methodology: Reads clearly, but more information required about how participants were recruited and sampled.

Response 2: Thank you very much for your comment. We provided additional information about how to get the representativeness of the baseline and how the panel vendor encouraged the participant to answer the surveys. Changes have been made in Participants section.

Point 3: How were the four groups (healthy etc) defined?

Response 3: Thank you very much for your contribution. The four analysed groups were defined depending on whether was established as positive or negative at baseline and during the follow-up. Then, four groups were categorized: Healthy; Persistence; Incidence; and Recovery. The definition of each group was informed in the Objectives, Variables & Instruments section, and the Conclusions, but following your suggestions, we made some changes at the Objective section to improve clarity, and we added the definition of each group at the first paragraph of the Discussion section as a reminder.

Point 4: Make sure to write "data were", not "data was".

Response 4: Thank you for the contribution. “Data was” has been changed by “data were”.

Point 5: Discussion: Make sure to define the four groups of the study to strengthen argument and connections. Limitations section should be put at the end of the discussion as it disrupts the flow of the discussion (comparison to other studies etc) with it in the middle.

Response 5: Thanks for your recommendation. At the beginning of the Discussion, the four groups analysed have also been defined as a reminder and in order to strengthen argument and connections during the Discussion. Limitations & Strengths section has been moved to the end of the Discussion.

Reviewer 2 Report

This paper aims to investigate the relationship between resilience and mental health during the covid-19 pandemic. Specifically, it seeks to delve into low resilience as a risk factor for the occurrence of mental health problems in the population. 

As a starting point, this seems a fairly simple premise given that the protective role of resilience in the face of stress and anxiety is well known, not only from studies conducted prior to the advent of covid-19, but also from the number of studies conducted since 2020.

However, as the reading progresses, an appropriate theoretical approach can be appreciated, which tries to combine its role in disorders such as MDE, GAD, STS and PTSD. 

The longitudinal design (and in this sense it is novel with respect to a large majority of cross-sectional studies published so far) is of special interest, since it allows a more solid approach to the effects of the pandemic on the population.   

The statistical analyses seem adequate, especially considering the incorporation of new subjects and dimensions throughout the study.

The results are clear and well presented. Regarding the discussion, I found interesting the way it is presented under headings that highlight the main findings of the study, its limitations and strengths, and the comparison with other studies, among others. 

 Having said all this, I would like to highlight some aspects that I suggest the authors to change or deepen.

Firstly, the authors' interest has focused on certain variables, leaving aside, as in any study, others that did not fit into a given theoretical framework. In this case, I would point to stress, fear of the disease itself or family and sociodemographic factors that have influenced, as the literature indicates, the psychological response to the pandemic. In particular, the relationship between stress and resilience has been studied. Some reference to these factors, beyond some mention in the limitations of the study, would make the text more complete.

On the other hand, as indicated in the limitations, not all variables, and the most important in this study, resilience, were measured before the pandemic. While it is appropriate to note the assumption of construct stability by the measurement instrument, the results should be assumed with caution.

Finally, I think that the interpretation of the results obtained in the group that has suffered from covid-19 or compatible symptoms should be significantly improved. The possible brain damage caused by the disease, as raised in the conclusions, requires further explanation.

Author Response

Response to Reviewer 2 Comments

Point 1: Firstly, the authors' interest has focused on certain variables, leaving aside, as in any study, others that did not fit into a given theoretical framework. In this case, I would point to stress, fear of the disease itself or family and sociodemographic factors that have influenced, as the literature indicates, the psychological response to the pandemic. In particular, the relationship between stress and resilience has been studied. Some reference to these factors, beyond some mention in the limitations of the study, would make the text more complete.

Response 1: Thank you for your comment, we added a new paragraph referencing these factors considering as a risk or resilient factors in the Comparison with other studies section in the Discussion section.

Point 2: On the other hand, as indicated in the limitations, not all variables, and the most important in this study, resilience, were measured before the pandemic. While it is appropriate to note the assumption of construct stability by the measurement instrument, the results should be assumed with caution.

Response 2: Thank you for your comment, we have added a sentence in Limitations & Strenghts section about the consequences of assessed resilience during the follow-up instead of at baseline and considering that the resilience assessed should be considered as a trait which is stable over time highlighting that the results should be considered with caution.

Point 3: Finally, I think that the interpretation of the results obtained in the group that has suffered from covid-19 or compatible symptoms should be significantly improved. The possible brain damage caused by the disease, as raised in the conclusions, requires further explanation.

Response 3: Thank you for your comment. We added more information about how COVID-19 affects brain damage and what consequences it produces. Changes have been made in COVID-19 subgroup section in the Discussion.